# Temperature Distribution Characteristics of Concrete during Fire Occurrence in a Tunnel

**Seungwon Kim [1],\*** , **Jaewon Shim [2]** , **Ji Young Rhee [2]** , **Daegyun Jung [1]**
**and Cheolwoo Park [1],\***

[1] Department of Civil Engineering, Kangwon National University, 346, Jungang-ro, Samcheok-si 25913, Korea;
sso2247@gmail.com

[2] Research Institute, Korea Expressway Corporation, 208-96, Dongbu-daero, 922 beon-gil,
Hwaseong-si 18489, Korea; this2pass@ex.co.kr (J.S.); need@ex.co.kr (J.Y.R.)

\* Correspondence: inncoms@kangwon.ac.kr (S.K.); tigerpark@kangwon.ac.kr (C.P.);
Tel.: +82-33-570-6518 (S.K.); +82-33-570-6515 (C.P.)

**Abstract:** Fire in a tunnel or an underground structure is characterized by a rise in temperature above 1000 °C in 5–10 min, which is due to the characteristics of the closed space. The Permanent International Association of Road Congresses has reported that serious damage can occur in an underground structure as a consequence of high temperatures of up to 1400 °C when a fire accident involving a tank lorry occurs in an underground space. In these circumstances, it is difficult to approach the scene and extinguish the fire, and the result is often casualties and damage to facilities. When a concrete structure is exposed to a high temperature, spalling or dehydration occurs. As a result, the cross section of the structure is lost, and the structural stability declines to a great extent. Furthermore, the mechanical and thermal properties of concrete are degraded by the temperature hysteresis that occurs at high temperatures. Consequently, interest in the fire safety of underground structures, including tunnels, has steadily increased. This study conducted a fire simulation to analyze the effects of a fire caused by dangerous-goods vehicles on the tunnel structure. In addition, a fire exposure test of reinforced-concrete members was conducted using the Richtlinien für die Ausstattung und den Betrieb von Straßentunneln (RABT) fire curve, which is used to simulate a tunnel fire.

**Keywords:** RABT fire curve; fire simulation; tunnel fire; high temperature; fire safety; fire accident

## 1. Introduction

With the recent sharp increase in accidents involving transport vehicles carrying hazardous materials (e.g., explosive flammables), damage to highway infrastructure facilities, such as tunnels, has increased substantially [1,2]. Detailed summaries of road and rail tunnel fire events clearly show the importance of considering fire risk in the design of tunnels [3,4]. In particular, due to the extensions and elongations of tunnel structures and increasing passage access to popular town areas, it is urgently necessary to ensure the safety of tunnel structures against unexpected extreme disasters, such as fires [1,2]. For this reason, many developed countries are currently enhancing fire intensity standards and reviewing explosion resistance standards. In most countries, however, the assessment of, and response to, fire risks is still limited, and the maturity of the design goal is relatively low [1,2].

In 2001, two trucks collided in the Gotthard tunnel in Switzerland, resulting in a fire, as well as 11 deaths and many injuries [5]. The scale of the fire was approximately 120–200 MW, and the flame temperature was estimated at over 1000 °C [5]. The fire brigade experienced difficulty in accessing the fire scene for 48 h, causing damage over a length of 700 m inside the tunnel and spalling of up to 350 mm in depth [5]. The damage caused by this accident amounted to approximately $31 billion,

and restoration work lasted two months [5]. Even though the authorities were equipped with the latest disaster prevention facilities at the time of the accident, the damage was considerable, and this clearly demonstrates the importance of the response as well as of the preparation of operational facilities [2].

This study analyzed fire intensity and the effects of fire on tunnel structures in terms of depth by simulating fire occurrences resulting from accidents of tank lorries loaded with inflammables in a tunnel environment. Furthermore, in order to verify the validity of the numerical analysis model, the effects of geometrical elements on fire were examined by a fire exposure test on reinforced-concrete members using the Richtlinien für die Ausstattung und den Betrieb von Straßentunneln (RABT) fire curve

## 2. Characteristics of Fire in a Tunnel

### 2.1. Material Characteristics of Concrete Exposed to High Temperature

Exposure to high temperatures results in spalling or destruction of the coating on members due to the water vapor pressure created inside the concrete [2,6–8]. At 100–400 °C, $Al_2O_3$-, $Fe_2O_3$-, and tobermorite-based hydrates are dehydrated, resulting in the collapse of gel and cement hydrates. In moderate-strength concrete that has been exposed to high temperatures, voids are generated as the vapor inside the concrete evaporates at approximately 200 °C [2,6–8]. The deformation recovery ability is drastically lowered in this temperature range, and the elastic modulus decreases significantly at temperatures of over 600 °C [2,6–8].

Concrete exhibits a tendency toward decreasing compressive strength and elastic modulus when exposed to high temperatures. Concrete exposed to high temperatures causes cross-sectional defects due to surface peeling or scattering [2,6–8]. This phenomenon is called spalling. The main cause of this phenomenon is the water vapor pressure that is generated when the water inside the concrete expands in response to high heat [2,6–8].

It is known that when a concrete structure is exposed to temperatures of approximately 650 °C or higher, it loses 50% of its original strength. When it is exposed to temperatures of approximately 850 °C or higher, it loses its structural performance [2,6–8].

### 2.2. Temperature Distribution Characteristics Due to Vehicle Fire in a Tunnel

According to a report published by the Permanent International Association of Road Congresses on fire and smoke control in road tunnels [9], in the event of a fire in a tunnel, when air flows in through the tunnel entrance at 6 m/s, the temperature of the ceiling reaches approximately 400 °C from the spot of the fire to a point approximately 100 m from the tunnel exit [9]. According to Dutch regulations, when a fire occurs in a large tank lorry with a loading capacity of 50 $m^3$ or higher, the temperature rises to approximately 1400 °C.

## 3. Tunnel Fire Simulation

It is almost impossible to consider all possible fire situations involving dangerous-goods vehicles in experimental assessments and verifications for simulating fires that occur in a road-network system. Various costs and time-consuming limits exist in reality. Therefore, simulation analysis is typically used in such studies, and a real fire experiment is only conducted when necessary to complement the results of the simulation. Fire simulation is generally conducted through computational fluid dynamics (CFD) analysis. CFD analysis is actively applied to fire-modelling research at domestic and international facilities and to establish firefighting design and evacuation parameters [1]. It is the most important tool in fire engineering. Building upon the Field Model developed in the U.K., the National Institute of Standard and Technology (NIST) and the Building and Fire Research Laboratory in the United States have achieved continuous developments in this area since 2000 [1]. CFD analysis, in particular, can be used to examine the thermal-fluid flow phenomenon on a large scale, for which life-sized model experiments are impossible to conduct. It can also quantify the degree of damage,

such as the size of the fire, smoke generation, toxic gas generation, and amount of radiation heat [1]. Furthermore, simulation is possible above the normal test-performance limits and can produce results according to specific scenarios, which enables the quantitative assessment of fire risks.

### 3.1. Modelling

The Fire Dynamic Simulator (FDS, version 6.5.3) [10], which has been developed by the NIST in the U.S., is the numerical analysis model used for the fire analysis in this study. The target space of the analysis was 45 m × 8 m × 5.5 m (length × width × height), and the length of the tunnel was assumed to be 45 m. It was believed that the length of the tunnel would not be affected by the temperature of the fire. As shown in Figure 1, a box-type cross section was applied to mimic the cross-sectional shape of the tunnel. The fire source was located between the center of the tunnel and its side. Tank lorries of 27 m$^3$ (27,000 L) volume were used with diesel as the fuel. The open-boundary condition was applied at the tunnel entrance and exit. In addition, as in a real situation, the combustion rate method was used to simulate the fire in tank lorries.

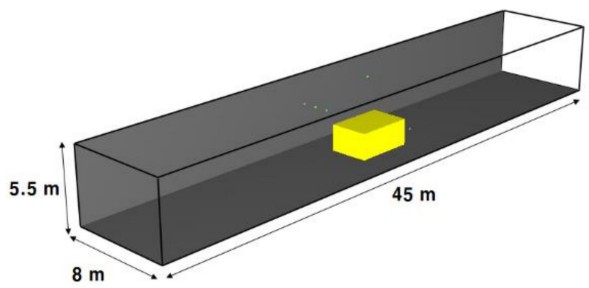

**Figure 1.** Dimension of tunnel and position of fire in tunnel modelling.

### 3.2. Analysis of Fire by Combustion Material Type

Comparative analyses were carried out on the changes in the fire characteristics according to the duration of ignition sources that were used to simulate fires for different materials in the FDS. The simulations were conducted by modelling for materials with different properties (e.g., combustion heat, density, and ignition point). Based on the results, the appropriate duration of the ignition source was determined and applied to the fire simulation.

Diesel and octane ($C_8H_{18}$), which are the commonly used gasoline types, as well as heptane ($C_7H_{16}$) and ethanol ($C_2H_6O$), were the hazardous materials applied to the fire simulation analysis. The values of the heat of combustion for these hazardous materials are 44.80 MJ/kg, 47.89 MJ/kg, 48.07 MJ/kg, and 29.65 MJ/kg, respectively. To analyze the changes in fire intensity with respect to the types of hazardous materials, specific analysis conditions were set to verify the fire intensity inside the tunnel when the fire occurred due to a hazardous material with a similar or lower heat of combustion. Table 1 outlines the model and conditions used for the analysis. Table 2 lists the material properties of the combustion materials.

**Table 1.** Analysis conditions by hazardous material type.

| Parameters | Input Data |
| --- | --- |
| Space size (m) | 45 × 8 × 5.5 |
| Grid size (m) | 0.5 × 0.5 × 0.5 |
| Conductivity of concrete (W/m·K) | 1.28 |
| Specific heat of concrete (KJ/kg·K) | 0.75 |
| Density of concrete (kg/m$^3$) | 2400 |
| Vehicle location | Middle of the tunnel 27,000 L capacity |
| Combustion material type | Diesel, ethanol, octane, and heptane |
| Simulation time (s) | 3600 |

**Table 2.** Properties of combustion materials considered in this study.

| Parameters | Diesel | Ethanol | Octane | Heptane |
|---|---|---|---|---|
| Density (kg/m$^3$) | 840 | 787 | 700 | 684 |
| Specific heat (kJ/kg·K) | 1.89 | 2.45 | 2.15 | 2.25 |
| Thermal conductivity (W/m·K) | 0.18 | 0.17 | 0.13 | 0.124 |
| Emissivity | 0.9 | 1.0 | 0.9 | 0.9 |
| Heat of combustion (kJ/kg) | 44,800 | 29,653 | 47,898 | 48,074 |
| Boiling point (°C) | 250 | 76 | 125 | 98.5 |

Figure 2 shows the analysis results for different combustion materials. Except for ethanol, whose heat of combustion is 29.65 MJ/kg, diesel, octane, and heptane have similar values for heat of combustion, i.e., 44.80 MJ/kg, 48.07 MJ/kg, and 47.89 MJ/kg, respectively. These three fuels had a similar fire intensity level of approximately 160 MW. However, the results for diesel showed a longer fire duration by approximately 500 s. In comparison to these three fuels, the heat of combustion of ethanol is approximately 60%, and this fuel also resulted in a lower fire intensity. However, the duration of its fire was longer than 1 h, much longer than the fire durations of the other combustion materials.

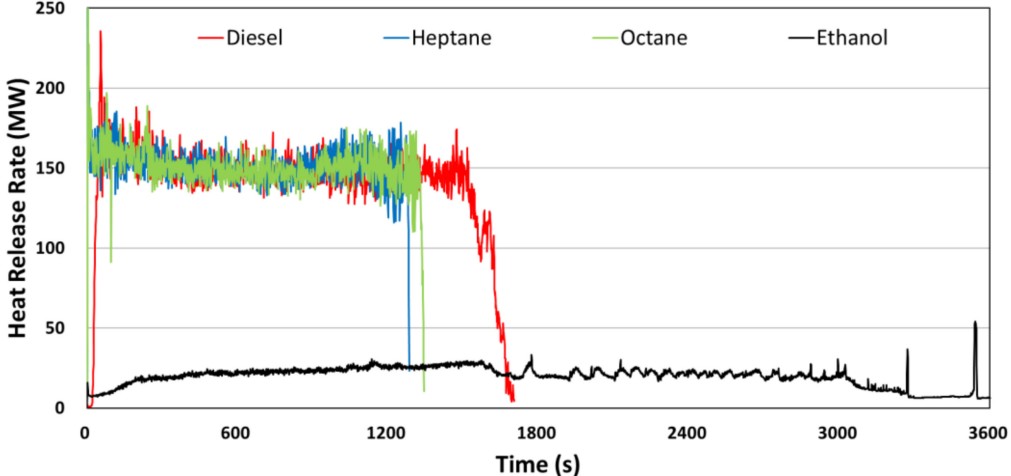

**Figure 2.** Heat release rates of the tested combustion materials.

The above analysis revealed that different types of hazardous materials on a highway can cause fires with different characteristics (i.e., the fire intensity and duration are likely to differ depending on the material type). The analysis of the simulation results showed that a fire caused by dangerous-goods vehicles carrying diesel results in the most serious conditions of fire intensity and fire duration.

*3.3. Analysis of Vehicle Fire in a Tunnel*

A simulation was conducted for a fire caused by dangerous-goods vehicles carrying diesel in a tunnel. The tunnel model used for this simulation was the same as that in Figure 2. Table 3 lists the analysis conditions used for this simulation.

**Table 3.** Vehicle fire analysis conditions.

| Parameters | Input Data |
| --- | --- |
| Space size (m) | $45 \times 8 \times 5.5$ |
| Grid size (m) | $0.25 \times 0.25 \times 0.25$ |
| Conductivity of concrete (W/m·K) | 1.28 |
| Specific heat of concrete (KJ/kg·K) | 0.75 |
| Density of concrete (kg/m³) | 2400 |
| Vehicle location | Middle of the tunnel 27,000 L |
| Measured depth of concrete wall (m) | Surface: 20, 40, 60, and 100 |
| Combustion material type | Diesel |
| Simulation time (s) | 3600 |

## 4. Fire Exposure Experiment on Reinforced-Concrete Member

In order to verify the analysis of the fire and heat-transfer characteristics of concrete in the event of a fire in a tunnel structure using the CFD analysis program FDS [10], the heat-transfer characteristics of a concrete member were examined through a fire experiment. The results were then compared with those of the CFD analysis. The fire exposure experiment was conducted using the RABT fire curve, which can simulate a tunnel fire for reinforced-concrete members.

### 4.1. RABT Fire Curve

The RABT fire curve was developed by the Road Construction Department of the German Ministry of Transportation under the Eureka Project [11]. In a simulated scenario, the temperature sharply rises to 1200 °C within 5 min after the beginning of the fire. The durations of fires involving trains and cars at a temperature of 1200 °C are 55 min and 25 min, respectively. The fire then cools down for 110 min. The RABT fire curve is known to have a shape similar to that of a real tunnel fire [11]. Figure 3 shows the RABT fire curves for railways and highways.

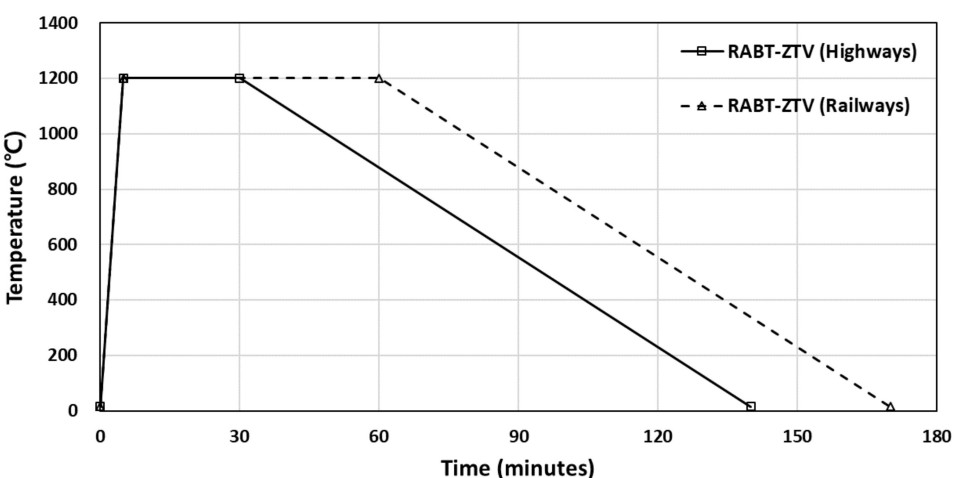

**Figure 3.** Richtlinien für die Ausstattung und den Betrieb von Straßentunneln (RABT) fire curves of highways and railways [11].

### 4.2. Dimensions of the Specimen and Experimental Method

The arrangement of reinforcement bars used in the RABT fire curve fire exposure experiment is the same as the arrangement in real road tunnels. Furthermore, Figure 4 shows the horizontal heating furnace for the high-temperature test used in the fire exposure experiment. Only the bottom surface of the specimen was exposed in this experiment. Ceramic fibers that can endure temperatures of up to approximately 1400 °C were installed to provide insulation between the top of the furnace and

the specimen. The horizontal heating furnace was designed to allow installation of a rectangular specimen with size of 1400 mm (length) × 1000 mm (width). The actual heating area was 1100 mm (length) × 700 mm (width). The design standard compressive strength of the concrete specimen that was used in this experiment was 27 MPa. Figure 5 shows the experimental setup for the fire exposure test using the RABT fire curve.

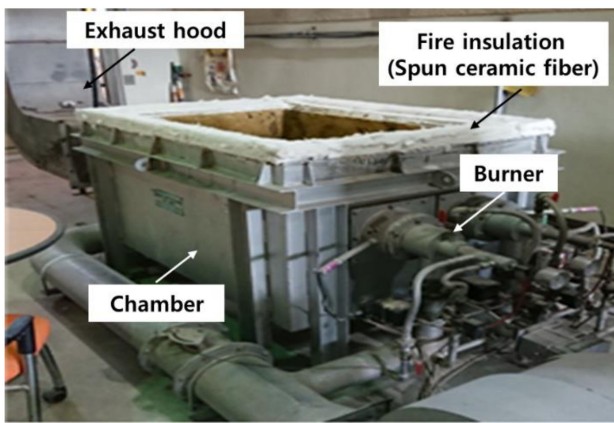

**Figure 4.** Horizontal heating furnace for the high-temperature test.

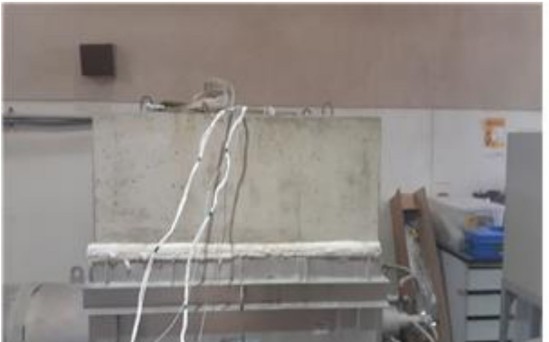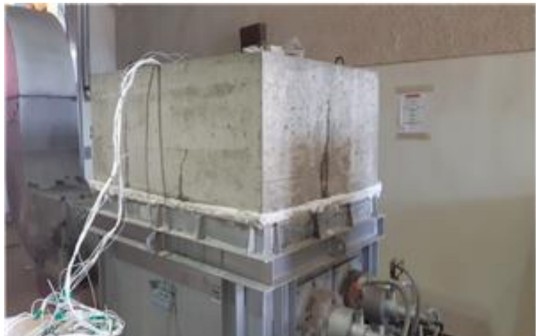

**Figure 5.** Experimental setup for the fire exposure test using the RABT fire curve.

To examine the heat-transfer characteristics using the RABT fire exposure experiment, thermocouples were installed at the center at 0, 20, 40, 60, 80, and 100 mm from the heating surface. Table 4 lists the positions of the thermocouples.

**Table 4.** Positions of the thermocouples.

| Thermocouples | | | Depth (mm) |
|---|---|---|---|
| **Left** | **Centre** | **Right** | |
| - | TC(C)-7 | - | 0 |
| TC(L)-1 | - | TC(R)-1 | 20 |
| TC(L)-2 | - | TC(R)-2 | 40 |
| TC(L)-3 | TC(C)-8 | TC(R)-3 | 60 |
| TC(L)-4 | - | TC(R)-4 | 80 |
| TC(L)-5 | - | TC(R)-5 | 100 |
| TC(L)-6 | - | TC(R)-6 | 120 |

## 5. Experimental Results and Analysis

### 5.1. Analysis of Vehicle Fire in the Tunnel

When a fire source corresponding to 27,000 L of diesel was located at the center of the tunnel, the fire intensity was approximately 150 MW, as shown in Figure 6. Figure 7 shows the results of the temperature measurements on the concrete surface and at depths of 20, 40, 60, and 100 mm. The analysis results for the concrete hydrothermal temperatures at each depth showed that for a fire of such a scale (27,000 L), the surface temperature rose to approximately 1000 °C at approximately 300 s after the occurrence of the fire. This high temperature was maintained for approximately 1800 s.

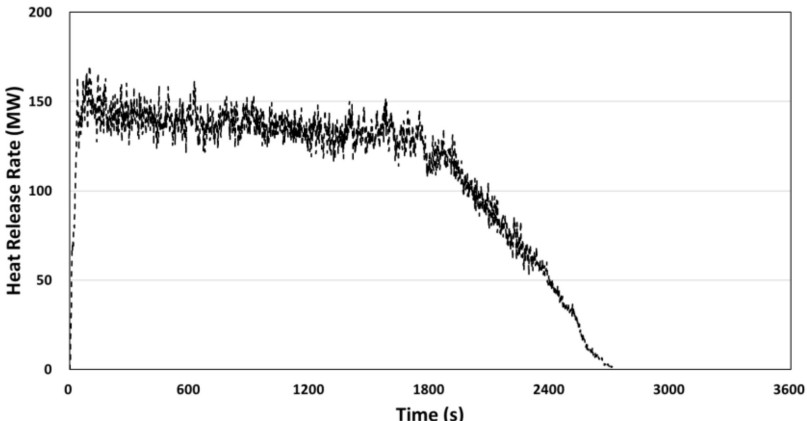

**Figure 6.** Fire intensity based on vehicle fire analysis.

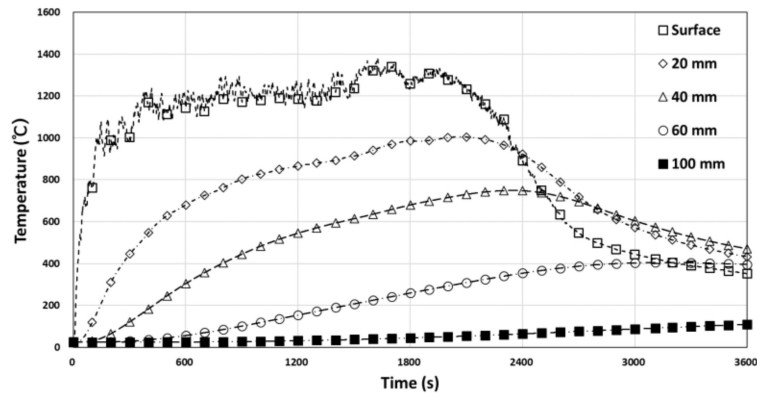

**Figure 7.** Temperature distribution by depth based on vehicle fire analysis.

The peak temperature of the concrete surface was approximately 1380 °C. The corresponding temperatures at depths of 20, 40, 60, and 100 mm were 1005 °C, 750 °C, 405 °C, and 110 °C, respectively. The International Tunneling Association (ITA) [12] specifies that the maximum critical temperatures for concrete and reinforcement bars should be 380 °C and 250 °C, respectively. On the basis of these results, the maximum critical temperature standard could not be met at concrete depths higher than 60 mm.

Figure 8 shows the temperature distribution inside the tunnel. Furthermore, the fire exposure experiment was conducted for real concrete members by applying the RABT fire curve where the temperature rose sharply to 1200 °C within 5 min. The result was similar to the that of the empirical verification of the tunnel structures' characteristics after exposure to high temperatures [12].

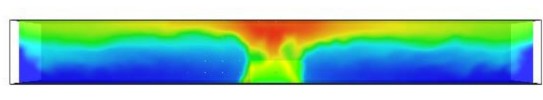
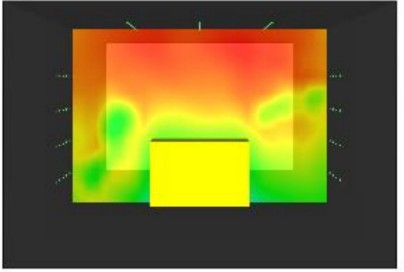

(**a**) Temperature distribution inside the tunnel (floor-plan view)     (**b**) Temperature distribution inside the tunnel (cross-sectional view)

**Figure 8.** Temperature distribution inside the tunnel.

## 5.2. Fire Exposure Experiment of Reinforced-Concrete Member

Figures 9 and 10 compare the specimen surface before and after the fire test based on RABT-ZTV (highways) 30 min fire curves and RABT-ZTV (railways) 60 min fire curves. Spalling is a complex process, which occurred in the concrete specimen due to the rapid temperature increase in the furnace. On the basis of the temperature changes in Figures 11 and 12, the result for the left thermocouple shows an abnormal pattern of temperature change with depth. On the basis of the right thermocouple, section loss appeared from the heating surface to 60 mm, and no section loss occurred above 80 mm. The temperature inside the concrete increased sharply depending on the generation of spalling. The temperature measured at approximately 60 mm, where the spalling occurred, was similar to that inside the furnace. At a depth of 80 mm, the temperature was approximately 420 °C during the rising period and approximately 540 °C during the descending period. However, some differences were evident because the spalling by heat varied depending on the condition of the concrete heating surface. The range of section loss can also be seen in Figures 9 and 10, which show the heating surface after the fire test was completed. Considering the arrangement depth of the reinforcement bars inside the specimen, the section loss was found to be 60–80 mm.

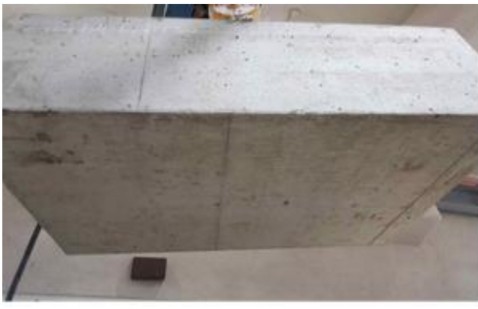
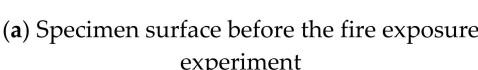
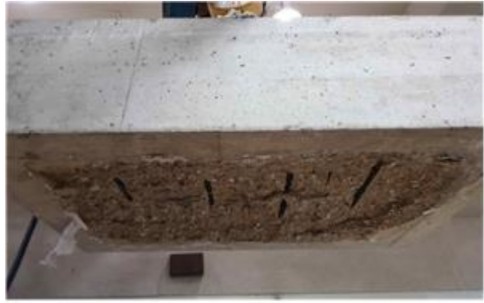

(**a**) Specimen surface before the fire exposure experiment     (**b**) Specimen surface after the fire exposure experiment

**Figure 9.** Specimen surface before and after the RABT-ZTV (highways) fire experiment.

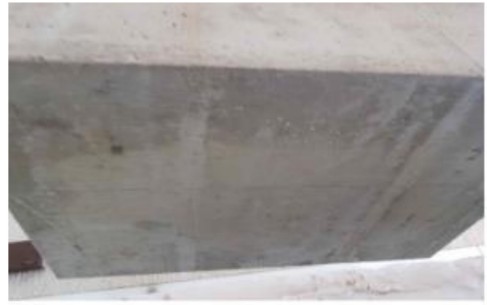

(**a**) Specimen surface before the fire exposure experiment

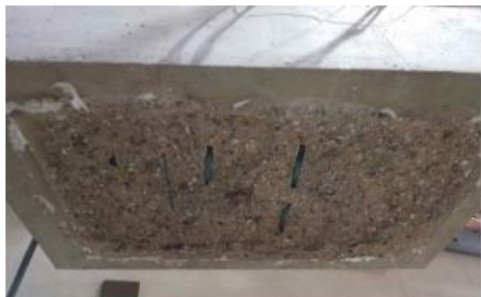

(**b**) Specimen surface after the fire exposure experiment

**Figure 10.** Specimen surface before and after the RABT-ZTV (railways) fire experiment.

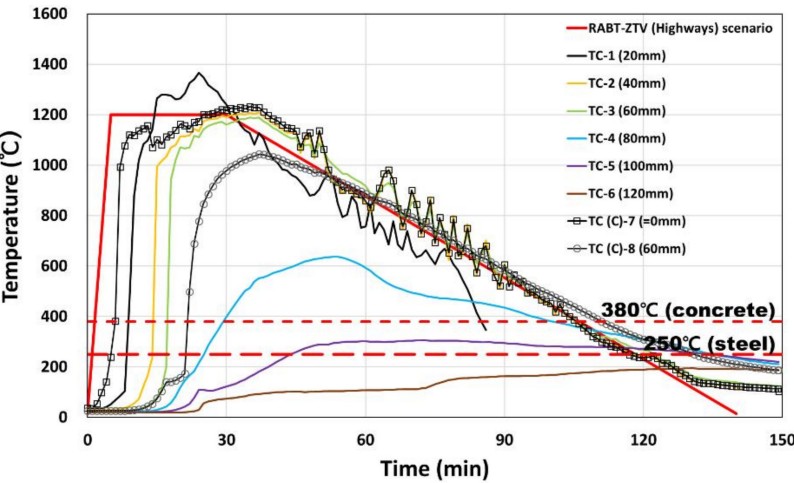

**Figure 11.** Temperature measurement results for the RABT-ZTV (highways) fire experiment.

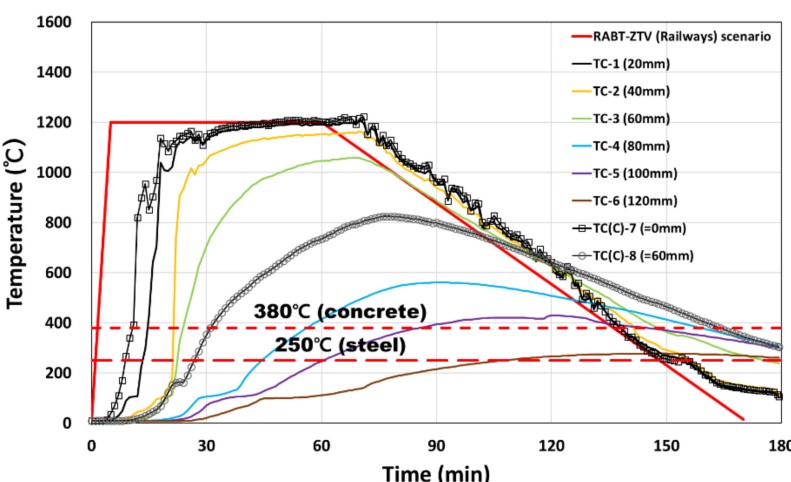

**Figure 12.** Temperature measurement results for the RABT-ZTV (railways) fire experiment.

On the basis of the results of the fire experiments, the appropriate coating thickness of the tunnel structure should be at least 80 mm in order to protect the concrete and internal reinforcement bars from fire. This contradicts the the simulation result that assumed that there had been no spalling in real concrete.

## 6. Conclusions

This study analyzed the effects of fire in a tunnel due to an accident involving dangerous-goods vehicles using fire simulation. Furthermore, a fire exposure test of a reinforced-concrete member was conducted using the RABT fire curve, which is employed to simulate a tunnel fire. The conclusions of this study are as follows:

(1) The results of the FDS tunnel fire analysis for each combustion material showed that diesel had a longer fire duration at the same fire intensity in comparison to the other tested fuels. As diesel is also the most commonly used fuel, it was selected as the fire source in the fire analysis that followed.

(2) The results of the fire analysis showed that the peak temperature on the concrete surface was approximately 1380 °C and that at a depth of 60 mm, the temperature was approximately 405 °C. These values do not satisfy the maximum critical temperatures for concrete and reinforcement bars that have been suggested by the ITA. These maximum temperatures are 380 °C and 250 °C, respectively. When a fire corresponding to 27,000 L of diesel occurs, the desirable coating thickness of the tunnel structure is 80 mm, which is greater than 60 mm.

(3) The analysis results for fire caused by dangerous-goods vehicles in the tunnel showed that at approximately 300 s after the beginning of the fire, the surface temperature of the concrete increased to approximately 1000 °C. This high temperature was maintained for approximately 1800 s. Therefore, on the basis of the empirical verification of the high-temperature exposure characteristics of concrete structures, the RABT fire curve was applied, in which the temperature rapidly increased to 1200 °C within 5 min. These results were similar to those of the analysis. Furthermore, since the maximum temperatures for concrete and reinforcement bars are specified as 380 °C and 250 °C, respectively, by the ITA, it is desirable to determine the fire damage that has been caused to the specimen on the basis of the suggested standards when an RABT fire curve with a significant temperature rise is used.

(4) The results of the RABT fire experiment showed serious spalling to a depth of 60 mm, or, in other words, the point at which reinforcement bars are located in concrete. The temperature inside the concrete rose rapidly depending on the spalling. The temperature measured at a depth of approximately 60 mm, where spalling occurred, was similar to that inside the furnace. The temperature at a depth of 80 mm was approximately 420 °C during the rising period and 540 °C during the descending period.

(5) On the basis of the results of the RABT fire curve experiments, 80 mm is the desirable minimum coating thickness of a tunnel concrete member in order to secure the safety of such structures during serious fires.

**Author Contributions:** Conceptualization, S.K., J.S., J.Y.R., C.P.; methodology, S.K., D.J.; writing and draft preparation, S.K., C.P.

**Funding:** This research was supported by the Basic Science Research Program through the National Research Foundation of Korea (NRF) and funded by the Ministry of Education (grant no. 2017R1A2B4012678) and the Korea Expressway Corporation Research Institute as part of the research project "Simulation and Experimental Simulation for verification of fire and explosion safety measures for vehicle fires".

**Conflicts of Interest:** The authors declare no conflicts of interest.

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
