# Peer review of "Temperature Distribution Characteristics of Concrete during Fire Occurrence in a Tunnel"

_applsci, doi:10.3390/app9224740_

Round 1

Reviewer 1 Report

This article can be published after performing the following modifications: 

The Introduction section needs to be modified exclusively. Figure 2, needs to be modified, and the text are cropped. All Tables need to be in the same format, Table 3 has an unnecessary line in the middle. In generally, the resolution of the photos are not good, e.g. Figure 13, Figure 8(left) Please take care of the typos, and errata throughout the paper. Also authors are advised to add more relevant references such as:

Schrefler, Bernhard & Brunello, Pierfrancesco & Gawin, Dariusz & Majorana, Carmelo & Pesavento, Francesco. (2002). Concrete at high temperature with application to tunnel fire. Computational Mechanics. 29. 43-51. doi: 10.1007/s00466-002-0318-y.

Author Response

Point 1:

The Introduction section needs to be modified exclusively.

Response 1:

Thank you for your comments.

Introduction part has been modified.

Point 2:

Figure 2, needs to be modified, and the text are cropped.

Response 2:

Thank you for your comments.

Figure 2 is modified.

Point 3:

All Tables need to be in the same format, Table 3 has an unnecessary line in the middle.

Response 3:

Thank you for your comments.

All tables are modified in the same format.

Point 4:

In generally, the resolution of the photos are not good, e.g. Figure 13, Figure 8(left) Please take care of the typos, and errata throughout the paper.

Response 4:

Thank you for your comments.

Figure 13 is not in the paper, and figure 8 (left) was captured from FDS. The resolution of the figure cannot be improved.

Point 5:

Also authors are advised to add more relevant references such as:

Schrefler, Bernhard & Brunello, Pierfrancesco & Gawin, Dariusz & Majorana, Carmelo & Pesavento, Francesco. (2002). Concrete at high temperature with application to tunnel fire. Computational Mechanics. 29. 43-51. doi: 10.1007/s00466-002-0318-y.

Response 5:

Thank you for your comments.

I add more relevant references.

Reviewer 2 Report

The authors discussed on the effects of the temperature distribution on concrete when a fire accident occurs in a tunnel. The problem has been faced in both way numerical and experimental. The results are satisfactory and in good agreement between numerical and experimental ones.
The manuscript is well written and organized (even if the english writing should be revised). In any case, for full acceptance of pubblication, some issues listed below should be faced.

1) Specify the acronysm introduced in the abstract: RABT fire curve?

2) The definition of the figure 1, 2 and 3 is very poor. They seem taken from
an image. Try to improve them especially with regards the writing. Moreover,
the figures 2 and 3 could be put together in a unique figure.

3) Figure 4: insert some writing related to the legend and specify better what
represent the two parallelepipeds shown in the figure. Also the label could be
written in more extesive way ("Tunnel modelling" is a little generic).

4) In general, in a scientific paper, it is not usual write in first plural
person (i.e. line 101 "We applied...", line 118 "We selected..." and so on),
try to invert the sentences writing in third singular person.

5) Figure 10: How have the temperatures been measured along the different
thicknesses of the concrete?

6) Figure 14, cited in the line 208, doesn't exist. Maybe it was an numeration
error.

7) Paragraph 5.2: in this paragraph have been described four figures. For this
reason, they would deserve a better and detailed comment.

Author Response

Point 1:

Specify the acronysm introduced in the abstract: RABT fire curve?

Response 1:

Thank you for your comments.

RABT (Richtlinien für die Ausstattung und den Betrieb von Straßentunneln) is acronized in abstract.

Point 2:

The definition of the figure 1, 2 and 3 is very poor. They seem taken from an image. Try to improve them especially with regards the writing. Moreover, the figures 2 and 3 could be put together in a unique figure.

Response 2:

Thank you for your comments.

Definition of the figure 1 and 3 has been modified and figure 2 is modified.

Point 3:

Figure 4: insert some writing related to the legend and specify better what represent the two parallelepipeds shown in the figure. Also the label could be written in more extesive way ("Tunnel modelling" is a little generic).

Response 3:

Thank you for your comments.

In figure 4, parallelepipeds are burner and is mentioned in figure too.

Point 4:

In general, in a scientific paper, it is not usual write in first plural person (i.e. line 101 "We applied...", line 118 "We selected..." and so on), try to invert the sentences writing in third singular person.

Response 4:

Thank you for your comments.

Sentences have been corrected.

Point 5:

Figure 10: How have the temperatures been measured along the different thicknesses of the concrete?

Response 5:

Thank you for your comments.

It was measured by thermocouples which is also mentioned at table 4.

Point 6:

Figure 14, cited in the line 208, doesn't exist. Maybe it was an numeration
error.

Response 6:

Thank you for your comments.

I have not cited figure 14.

Point 7:

Paragraph 5.2: in this paragraph have been described four figures. For this reason, they would deserve a better and detailed comment.

Response 7:

Thank you for your comments.

It is modified.

Round 2

Reviewer 1 Report

The work was improved based on my suggestions and recommendations.